# Altered Transmission of Cardiac Cycles to Ductus Venosus Blood Flow in Fetal Growth Restriction: Why Ductus Venosus Reflects Fetal Circulatory Changes More Precisely

**DOI:** 10.3390/diagnostics12061393

**Published:** 2022-06-04

**Authors:** Naomi Seo, Yasushi Kurihara, Tomoki Suekane, Natsuko Yokoi, Kayoko Nakagawa, Mie Tahara, Akihiro Hamuro, Takuya Misugi, Akemi Nakano, Masayasu Koyama, Daisuke Tachibana

**Affiliations:** 1Department of Obstetrics and Gynecology, Graduate School of Medicine, Osaka Metropolitan University, 1-4-3 Asahimachi Abeno-ku Osaka, Osaka 545-8585, Japan; kurikuri_1011@yahoo.co.jp (Y.K.); wadanatsuko@gmail.com (N.Y.); mtahara@med.osaka-cu.ac.jp (M.T.); hamuroa@med.osaka-cu.ac.jp (A.H.); misutaku1975@infoseek.jp (T.M.); akeake@med.osaka-cu.ac.jp (A.N.); masayasukoyama@gmail.com (M.K.); m1159899@med.osaka-cu.ac.jp (D.T.); 2Department of Gynecology, Saiseikai Senri Hospital Based On Social Welfare Organization Saiseikai Imperial Gift Foundation, 1-1-6 Tsukumodai, Suita-shi 565-0862, Japan; manchestercity0217@outlook.jp; 3Department of Obstetrics and Gynecology, Izumiotsu City Hospital, 16-1 Gezyocho Izumiotsu, Osaka 595-0027, Japan; kayopu_a@hotmail.com

**Keywords:** fetal growth restriction, ductus venosus, cardiac cycle, time-related analysis, Doppler

## Abstract

We aimed to investigate the relation between the time intervals of the flow velocity waveform of ductus venosus (DV-FVW) and cardiac cycles. We defined Delta A as the difference in the time measurements between DV-FVW and cardiac cycles on the assumption that the second peak of ductus venosus (D-wave) starts simultaneously with the opening of the mitral valve (MV). As well, we defined Delta B as the difference of the time measurements between DV-FVW and cardiac cycles on the assumption that the D-wave starts simultaneously with the closure of the aortic valve (AV). We then compared Delta A and Delta B in the control and fetal growth restriction (FGR) groups. In the control group of healthy fetuses, Delta A was strikingly shorter than Delta B. On the other hand, in all FGR cases, no difference was observed. The acceleration of the D-wave is suggested to be generated by the opening of the MV under normal fetal hemodynamics, whereas it precedes the opening of the MV in FGR. Our results indicate that the time interval of DV analysis might be a more informative parameter than the analysis of cardiac cycles.

## 1. Introduction

Fetal growth restriction (FGR) caused by placental dysfunction is a challenging disease, and the appropriate timing of delivery in FGR cases is still controversial because it requires a balance between the risks of prolonged exposure to hypoxemia, which may lead to fetal demise, and the risks of prematurity to adapt to dramatic changes in respiratory and cardiocirculatory circumstances [1,2,3,4]. Impaired oxygenation and insufficient return of umbilical venous blood flow lead to the redistribution of blood to the vital organs, and these phenomena are observed by the elevated pulsatility index (PI) in the umbilical and dilated middle cerebral arteries via Doppler ultrasonography.

In the progression of fetal deterioration, a Doppler evaluation of the fetal venous system, including ductus venosus (DV), plays a pivotal role in close fetal monitoring [5,6]. DV-PI has been widely evaluated by many researchers and is thought to be one of the most useful parameters by which to make a decision to deliver a decompensated fetus at the appropriate timing. The widely accepted theory of how the flow velocity waveform of DV (DV-FVW) is generated by cardiac cycles is thought to be as follows: the first peak (S-wave) corresponds to ventricular systole, and the second peak (D-wave) to ventricular diastole; the a-wave, which is lowest throughout the DV-FVW, is induced by atrial contraction. Recently, based on these unique features of the DV-FVW, we reported on the physiological changes occurring in the time intervals of systolic and diastolic components of DV-FVWs in normal fetuses in relation to fetal cardiac cycles [7]. We also depicted the alterations in time intervals of DV-FVWs in severe FGR fetuses, despite these alterations not being precisely mirrored by the cardiac cycles in pathophysiological conditions [8]. Furthermore, Sanapo et al. also reported that parameters of the DV-FVW do not necessarily reflect only cardiac functions in a study of fetuses at risk for cardiac dysfunction [9]. As such, parameters of the DV-FVW seem to be more valuable and useful for the evaluation of high-risk fetuses than those of cardiac cycles, although the mechanism remains to be clarified.

To elucidate these advantages of DV-FVW assessment, we aimed to conduct a meticulous investigation regarding the relation between the time intervals of DV-FVWs and fetal cardiac cycles, based on the clinical question as to whether the D-wave is generated at the timing of the opening of the atrioventricular valve.

## 2. Materials and Methods

### 2.1. Study Design, Ethical Approval, and Study Population

A cross-sectional study was performed on 60 normal fetuses, as a control group, from 25 to 33 gestational weeks at Osaka Metropolitan University Hospital, from December 2020 to February 2022. The fetuses showed normal morphology, and estimated fetal weights were within ±1.5 SD of the normal mean, according to local reference ranges, in all cases [10].

FGR fetuses (*n* = 23) were also retrospectively analyzed at both Osaka Metropolitan University Hospital and Osaka City General Hospital from April 2011 to November 2021. FGR was defined as an estimated fetal weight (EFW) < −2.0 SD of the local reference range and with an elevated umbilical artery PI > 95th percentile of the reference range [11]. Gestational age (GA) was calculated from the last menstrual period and confirmed by the crown-rump length measured between the 9th and the 11th gestational week. In the FGR group, only the last examination performed within 5 days of delivery was used retrospectively for analysis. Exclusion criteria for both the control group and the FGR group were multiple pregnancies, chromosomal abnormalities, or structural anomalies. We received informed written consent from all participants, and the study protocol was approved by the institutional review board of Osaka Metropolitan University Graduate School of Medicine on 1 October 2020 (Approved Number: 2020-183). Doppler studies were performed using a Voluson E8, E10 (GE Healthcare Ultrasound, Milwaukee, WI, USA) or a LOGIQ S6 (GE Healthcare Ultrasound) ultrasound machine with a transducer of 2.0–8.0 MHz or 2.0–5.5 MHz. The angle between the ultrasound beam and the direction of blood flow was <20°, and all recordings were taken without fetal breathing movements. DV was visualized in either a midsagittal or an oblique transverse section, and cases with poor signal quality were excluded from the study. The fetal heart rate was within the normal range of 120–160 bpm, and differences among measurements in each fetus were <5 bpm. In the measurement of DV-FVWs, the duration of each S-wave and D-wave was defined as (i) and (ii), respectively (upper figures in Figure 1a,b).

To evaluate the cardiac cycles, the Doppler sample volume was placed on the lateral wall of the ascending aorta, below the aortic valve (AV) and just above the mitral valve (MV) (lower figures in Figure 1a,b) [12]. In these settings, a Doppler trace provides both the MV and the AV movements so that the isovolumic contraction time (ICT), the ejection time (ET), the isovolumic relaxation time (IRT), and the myocardial performance index (MPI) can be measured [12].

Figure 1a shows our first hypothesis that the D-wave starts simultaneously with the opening of the MV, and Figure 1b shows our second hypothesis that the D-wave starts simultaneously with the closure of the AV. In the first hypothesis, we defined time intervals as follows: (iii) the duration from the top of the second peak of ventricular inflow (A-wave) to the opening of the MV and (iv) the duration from the opening of the MV to the top of the A-wave. In the second hypothesis, we defined time intervals as follows: (v) the duration from the top of the A-wave to the closure of the AV and (vi) the duration from the closure of the AV to the top of the A-wave.

Delta A was defined as the sum of each absolute value of the time difference between (i) and (iii) and that between (ii) and (iv); Delta B was defined the sum of each absolute value of the difference between (i) and (v) and that between (ii) and (vi).

### 2.2. Statistical Analysis

We compared Delta A and Delta B in the control and FGR groups. We also divided both groups into two sub-groups at examination: at ≤ 28 + 6 gestational age (GA) and >28 + 6 GA. We used the SPSS statistics version 20 for statistical analysis (SPSS Inc., Chicago, IL, USA). Comparisons between the control and FGR groups were performed using the Mann–Whitney U-test, and *p* < 0.05 was considered statistically significant.

## 3. Results

Out of the 45 FGR cases, 3 cases were excluded during the study period: 1 case had a ventricular septal defect that was diagnosed postnatally, 1 showed fetal tachycardia, and 1 case resulted in fetal demise. All parameters were obtained from 23 of the total FGR cases. The decision for delivery was made based on a non-reassuring fetal status judged by cardiotocography, and all FGR cases were delivered by cesarean section. Sixteen cases in the FGR group were included from a previous study [13]. Table 1 shows the maternal characteristics. Table 2 shows neonatal outcomes of the FGR group overall and according to GA at examination.

The measurements of cardiac parameters in the control and FGR groups are shown in Table 3. No difference was observed in each parameter between the two groups. Figure 2 shows the result of the difference between the two hypotheses. The results from the control group are shown in Figure 2a–c. Delta A was strikingly shorter than Delta B in all normal fetuses. On the other hand, no difference was observed in Delta A and Delta B in all FGR fetuses (Figure 2d–f). Furthermore, Delta A tended to increase in FGR fetuses ≤ 28 + 6 GA, although it did not reach significance (Figure 2e).

## 4. Discussion

### 4.1. Main Findings and Importance

The present finding that Delta A was apparently shorter than Delta B in normal fetuses suggests that the acceleration of the D-wave is generated simultaneously by the opening of the MV under normal fetal hemodynamics. In contrast, in the FGR group, no difference was observed between Delta A and Delta B, and this result suggests that the acceleration of the D-wave starts before the opening of the MV. Furthermore, in contrast, Delta A was significantly increased in those severe FGR cases which necessitated the early termination of pregnancy.

We previously demonstrated the systolic/diastolic time ratios of DV-FVWs and cardiac cycles, using similar methods as those in the FGR study, and the most significant decrease in systolic/diastolic ratios was observed in the DV-FVWs in severe FGR cases [8]. Furthermore, the systolic/diastolic time ratios of DV-FVWs showed a significant correlation with umbilical arterial pH at birth, as well as with the DV-PI, although the cardiac parameters did not show such a correlation [13]. These findings drove us to a further investigation in the present study; why is the fetal hemodynamic condition more discernably reflected in the time intervals of DV-FVWs than in the time-interval parameters of cardiac cycles?

To mention, theoretically, ventricular relaxation and high atrial pressure lead to the opening of the MV, thus accelerating venous forward velocities in the D-wave [8,14], and this theory was initially supported, by a mathematical method, in normal fetuses in the present study. On the other hand, we further revealed that the acceleration of the D-wave precedes the opening of the MV in severe FGR fetuses; in other words, the D-wave starts to be generated during the cardiac phase of isovolumetric relaxation. One possible explanation for this phenomenon might be that altered cardiac wall motion during isovolumetric relaxation, which is caused by chronic hypoxia/acidemia, functional hypovolemia, resulted from brain sparing (redistribution) [15,16,17], and changes in cardiac geometry [18], might force the pumping up of venous return under certain pathophysiological conditions in FGR. Furthermore, our test results of a relatively condensed hematocrit in the FGR group (median 49.7% (range 35.0% to 64.4%)), compared with normally grown neonates [19,20], also might have affected the viscosity of the fetal circulating blood. However, these factors did not affect cardiac parameters such as ICT, ET, IRT, and MPI.

### 4.2. Representative Alterations in Doppler Measurements of Ductus Venosus in FGR

Based on the findings in the present study, representative alterations in Doppler measurements of DV-FVWs and cardiac cycles are shown in Figure 3. Under normal physiological conditions, the D-wave of the DV-FVW starts at the timing of the opening of the MV (Figure 3a,b). In the pathophysiological circulation, on the other hand, the D-wave may be generated during IRT or just after the closure of the AV (Figure 3c,d).

Efforts for the understanding of complexities in multifactor-derived DV-FVWs have been undertaken by some investigators and from other perspectives. The nadir between the S-wave and the D-wave, the starting point of the acceleration of the D-wave, has been investigated via Doppler examination using blood flow velocity. Turan et al. studied the blood flow patterns of the DV-FVWs in fetuses at risk for cardiac dysfunction and speculated that the velocity of the nadir between the S-wave and the D-wave (they named this trough the ‘v-wave’) reflects ventricular relaxation and holodiastolic filling [21]. Later, Sanapo et al. (from the same Turan et al. group) reported that the individual velocity ratios of DV-FVWs have relationships with cardiac function which are not reflected by DV-PIs. Moreover, they emphasized that the assessment of the v-wave velocity may be valuable for the evaluation of individual events in cardiac cycles [9]. These reports from other groups using different methodologies might be in line with our findings that the pathological circulation in FGR may affect the cardiac wall motion during the isovolumetric relaxation period.

### 4.3. Study Limitations and Strengths

The limitations of the present study are that the data were retrospectively obtained from FGR fetuses and that it was not possible methodologically to obtain precise, simultaneous measurements of both DV-FVWs and cardiac parameters. Furthermore, the low statistical power might have had some effects on the results of the comparison of cardiac parameters such as ICT, ET, IRT, and MPI. The present study also lacks any speculation as for the pathogenesis of preeclampsia, which has substantial effects on fetal growth and placental function [22,23]. However, the strengths are that this is the first report to suggest a mechanism of the acceleration of the D-wave by measuring the time intervals of cardiac events using time-interval parameters and mathematical methods. We believe that this parameter might be valuable in the meticulous monitoring of fetuses at high risk for cardiac dysfunction. Longitudinal studies are needed to determine when this parameter becomes obvious in the process of fetal deterioration in FGR. Further explorations of the relationship between this parameter and clinical outcomes including neonatal circulation are expected.

## 5. Conclusions

We revealed that the acceleration of the D-wave may be generated simultaneously with the opening of the MV in physiological circulation, while, in contrast, the acceleration of the D-wave will start during the isovolumetric relaxation period in severely growth-restricted fetuses, thus suggesting that the alteration in time intervals of DV-FVWs will reflect not only cardiac cycles, but also pathophysiological systemic changes in FGR cases. We believe that the time interval analysis of DV-FVWs may provide a more comprehensive insight into the pathophysiological conditions of severe FGR fetuses.

## Figures and Tables

**Figure 1 diagnostics-12-01393-f001:**
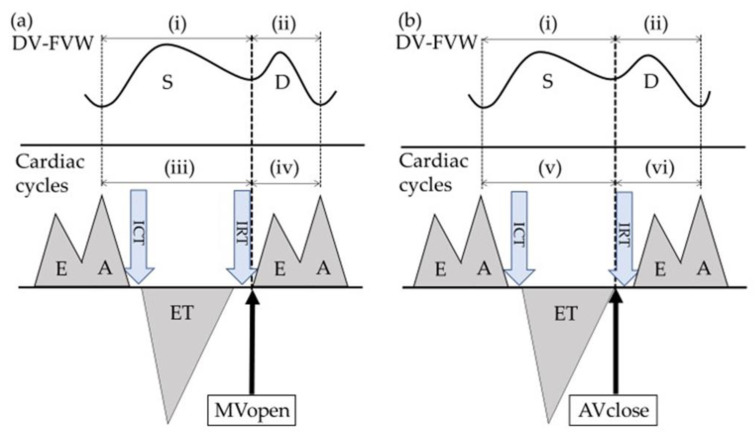
Schematization of the ductus venosus flow velocity waveform (DV-FVW) (upper figure) and cardiac cycles (lower figure). (**a**) Hypothesis based on the concept that the second peak of ductus venosus (D-wave) starts simultaneously with the opening of the mitral valve (MV). (**b**) shows the hypothesis based on the concept that the D-wave starts simultaneously with the closure of the aortic valve (AV). The isovolumic contraction time (ICT) was calculated from the closure of the MV to the opening of the AV, the ejection time (ET) was calculated from the opening to the closure of the AV, and the isovolumic relaxation time (IRT) was calculated from the closure of the AV to the opening of the MV. S-wave = the first peak of ductus venosus, E-wave = the first peak of ventricular inflow, A-wave = the second peak of ventricular inflow, MVopen = the timing of the opening of the MV, AVclose = the timing of the closure of the AV.

**Figure 2 diagnostics-12-01393-f002:**
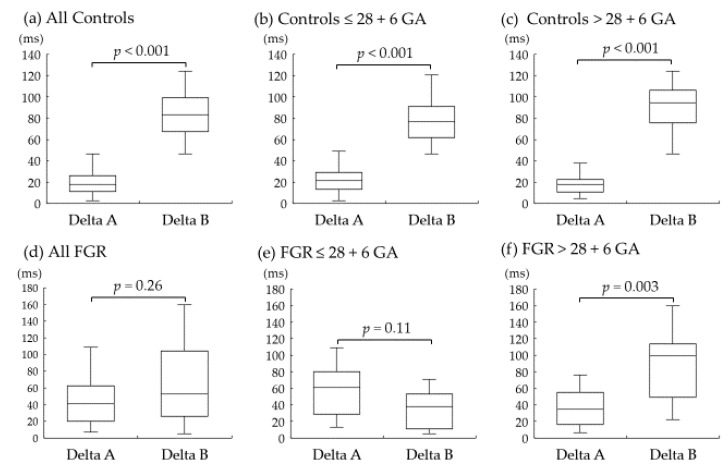
Box-and-whiskers plots showing Delta A and Delta B in the control and fetal growth restriction (FGR) groups overall and according to gestational age (GA) at examination. Delta A and Delta B are shown in all control fetuses (**a**), in control fetuses ≤ 28 + 6 GA (**b**), and in control fetuses > 28 + 6 GA at examination (**c**). Delta A and Delta B are shown in all FGR fetuses (**d**), in FGR fetuses ≤ 28 + 6 GA (**e**), and in FGR fetuses > 28 + 6 GA at examination (**f**). ms = millisecond.

**Figure 3 diagnostics-12-01393-f003:**
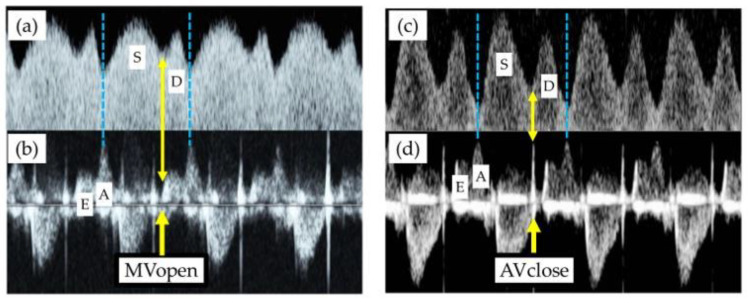
Representative alterations in Doppler measurements of ductus venosus flow velocity waveforms (DV-FVWs) (**a**,**c**) and cardiac cycles (**b**,**d**). (**a**,**b**) Normal physiological conditions. (**c**,**d**) Pathophysiological circulation. S-wave = the first peak of ductus venosus, D-wave = the second peak of ductus venosus, E-wave = the first peak of ventricular inflow, A-wave = the second peak of ventricular inflow, MVopen = the timing of the opening of the mitral valve, AVclose = the timing of the closure of the aortic valve.

**Table 1 diagnostics-12-01393-t001:** Maternal characteristics of women with uncomplicated pregnancy (controls) and women with pregnancy complicated by fetal growth restriction (FGR).

Characteristic	Controls (*n* = 60)	FGR (*n* = 23)	*p*
Age (years)	32.0 (18 to 44)	33.0 (22 to 44)	0.61
Nulliparous/parous	30/30	17/6	0.05
GA (weeks)	28.9 (25.0 to 33.6)	29.0 (26.0 to 33.4)	0.98

Data are given as median (range). GA = gestational age.

**Table 2 diagnostics-12-01393-t002:** Neonatal outcome of fetuses with fetal growth restriction (FGR) overall and according to gestational age (GA) at examination.

		GA (Weeks) at Examination:	
Outcome	All FGR (*n* = 23)	≤28 + 6 (*n* = 11)	>28 + 6 (*n* =12)
GA at delivery (weeks)	29.1 (26.1 to 33.9)	27.6 (26.1 to 29.1)	30.6 (29.0 to 33.9)
Measurement before delivery (days)	2.0 (0 to 5.0)	1.0 (0 to 5.0)	2.0 (0 to 4.0)
Birth weight (g)	726 (328 to 1256)	502 (328 to 760)	859 (576 to 1256)
1-min Apgar score	5 (1 to 9)	5 (1 to 7)	6 (3 to 9)
5-min Apgar score	8 (3 to 9)	6 (3 to 9)	8 (6 to 9)
Umbilical artery pH	7.25 (6.99 to 7.33)	7.25 (6.99 to 7.32)	7.24 (7.03 to 7.33)
Umbilical artery base excess	−4.1 (−14.5 to 2.3)	−3.4 (−11.0 to 2.3)	−4.8 (−14.5 to 0.9)
Neonatal hematocrit (%)	49.7 (35.0 to 64.4)	49.3 (35.0 to 55.2)	55.4 (44.2 to 64.4)

Data are given as median (range).

**Table 3 diagnostics-12-01393-t003:** Examined parameters in control and fetal growth restriction (FGR) fetuses.

Parameter	Controls (*n* = 60)	FGR (*n* = 23)	*p*
ICT (ms)	31.1 (20.0 to 57.8)	31.1 (20.0 to 42.2)	0.68
ET (ms)	165.6 (144.4 to 188.9)	166.7 (140.0 to 191.1)	0.81
IRT (ms)	44.4 (31.1 to 55.6)	46.7 (28.9 to 75.2)	0.39
MPI	0.47 (0.29 to 0.66)	0.49 (0.30 to 0.69)	0.58

Data are given as median (range). ICT = isovolumic contraction time, ET = ejection time, IRT = isovolumic relaxation time, MPI = myocardial performance index. ms = millisecond.

## Data Availability

All data related to this study are contained within the manuscript.

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
