# Peer review of "Altered Transmission of Cardiac Cycles to Ductus Venosus Blood Flow in Fetal Growth Restriction: Why Ductus Venosus Reflects Fetal Circulatory Changes More Precisely"

_diagnostics, 2022, doi:10.3390/diagnostics12061393_

Round 1
Reviewer 1 Report
I have read this paper with great interest, and applaud the efforts and construct of the authors.
In essence, the authors put forward the concept to assess Delta A and Delta B as an ‘sensitive parameter’. I agree that based on the current study, arguments have been generated to further study this approach, but I somewhat disagree on the use of ‘sensitivity’ as the concept has until current been assessed in a predefined control and FGR group. I therefore recommend to rephrase this.
If I understand the concept well, it is likely also better to use ‘shorter (in time)’ instead of smaller ?
I’m aware that timing of delivery in FGR cases remains controversial, but to further improve this, we likely need more ‘specific’ diagnostic tools, instead of ‘sensitive’ tools, as these are more suited for screening purposes. Related to this, do the authors put this diagnostic tool’ forward for screening, or for confirmation/quantification. Pending on this, the introduction should perhaps be somewhat rephrased, while this may also affect ‘diagnostic tool’ development (next steps to be taken). The opinion of the authors on these next steps should be added to the discussion: how should we use and further explore this ‘biomarker’.
Methods
Can you reflect on how you have handled repeated/multiple measurements/ultrasounds in the same case ?
Author Response
Dear Reviewer 1,
We appreciate your kind advice and thoughtful comments for refining our manuscript. Point-by-point answers are as follows, and added sentences in the manuscript are indicated by underlined text.
Comment 1.
I have read this paper with great interest, and applaud the efforts and construct of the authors. In essence, the authors put forward the concept to assess Delta A and Delta B as an ‘sensitive parameter’. I agree that based on the current study, arguments have been generated to further study this approach, but I somewhat disagree on the use of ‘sensitivity’ as the concept has until current been assessed in a predefined control and FGR group. I therefore recommend to rephrase this.
Answer 1.
We rephrased ‘informative’ instead of ‘sensitive’ in Page1, Line 23, ‘valuable’ instead of ‘sensitive’ in Page 2, Line 51 and ‘discernably’ instead of ‘sensitively’ in Page 5, Line 183 in the manuscript.
Comment 2.
If I understand the concept well, it is likely also better to use ‘shorter (in time)’ instead of smaller?
Answer 2.
I rephrased ‘shorter’ instead of ‘smaller’ in Page 1, Line 19, Page 4, Line 144 and Page 5, Line 170 in the manuscript.
Comment 3.
I’m aware that timing of delivery in FGR cases remains controversial, but to further improve this, we likely need more ‘specific’ diagnostic tools, instead of ‘sensitive’ tools, as these are more suited for screening purposes. Related to this, do the authors put this diagnostic tool’ forward for screening, or for confirmation/quantification. Pending on this, the introduction should perhaps be somewhat rephrased, while this may also affect ‘diagnostic tool’ development (next steps to be taken). The opinion of the authors on these next steps should be added to the discussion: how should we use and further explore this ‘biomarker’.
Answer 3.
We appreciate your thoughtful comments and advices. As we answered above, we rephrased ‘informative’ instead of ‘sensitive’ in Page1, Line 23, ‘valuable’ instead of ‘sensitive’ in Page 2, Line 51 and ‘discernably’ instead of ‘sensitively’ in Page 5, Line 183 in the manuscript. We have added the sentences, as follows, in discussion; We believe that this parameter might be valuable in the meticulous monitoring of fetuses at high risks. Longitudinal studies are needed to study when this parameter becomes obvious in the process of fetal deterioration in FGR. Further explorations of the relationship between this parameter and clinical outcomes including neonatal circulation are expected.
Comment 4.
Methods
Can you reflect on how you have handled repeated/multiple measurements/ultrasounds in the same case?
Answer 4.
We repeated these measurements for each parameter at least three times in a case.
We appreciate your kind consideration and thoughtful advice for publishing our findings.
June 1, 2022
Sincerely,
Naomi Seo
Department of Obstetrics and Gynecology, Osaka City University Graduate School of Medicine, 1-4-3 Asahimachi Abeno-ku Osaka, Osaka, Japan 545-8585
E-mail: d19mb027@gmail.com
Tel: +81 6 6645 3862
Fax: +81 6 6646 5800
Reviewer 2 Report
Dear authors,
I read with great interest the manuscript, which falls within the aim of this Journal.
the DV acts as a bypass of the liver microcirculation and plays a critical role in the fetal circulation. The DV allows oxygenated and nutrient-rich venous blood to flow from the placenta to the myocardium and brain. Increased impedance to flow in the fetal DV is associated with fetal aneuploidies, cardiac defects and other adverse pregnancy outcomes.
The time interval analysis of DV-FVWs may provide a more comprehensive insight into the pathophysiological conditions in severe FGR fetuses.
In my honest opinion, the topic is interesting enough to attract the readers’ attention. Nevertheless, authors should clarify some points and improve the discussion, as suggested below. Authors should consider the following recommendations:
You can focus on the possible consequencies of a preterm birth and about the management of postpartum hemorrhage that can happens in these pts as well the possible management of preclampsia .
I suggest you to read anc cite these papers.
The role of serum potassium and sodium levels in the development of postpartum hemorrhage. A retrospective study
The role of endoglin and its soluble form in pathogenesis of preeclampsia
Correlation between maternal gingivitis/periodontitis and preterm delivery: Fact or fancy?
Author Response
Dear Reviewer 2,
We appreciate your kind advice and thoughtful comments for refining our manuscript. Point-by-point answers are as follows, and added sentences in the manuscript are indicated by underlined text.
Comment 1.
Dear authors,
I read with great interest the manuscript, which falls within the aim of this Journal.
the DV acts as a bypass of the liver microcirculation and plays a critical role in the fetal circulation. The DV allows oxygenated and nutrient-rich venous blood to flow from the placenta to the myocardium and brain. Increased impedance to flow in the fetal DV is associated with fetal aneuploidies, cardiac defects and other adverse pregnancy outcomes. The time interval analysis of DV-FVWs may provide a more comprehensive insight into the pathophysiological conditions in severe FGR fetuses. In my honest opinion, the topic is interesting enough to attract the readers’ attention. Nevertheless, authors should clarify some points and improve the discussion, as suggested below. Authors should consider the following recommendations: You can focus on the possible consequencies of a preterm birth and about the management of postpartum hemorrhage that can happens in these pts as well the possible management of preeclampsia. I suggest you to read anc cite these papers. The role of serum potassium and sodium levels in the development of postpartum hemorrhage. A retrospective study
The role of endoglin and its soluble form in pathogenesis of preeclampsia
Correlation between maternal gingivitis/periodontitis and preterm delivery: Fact or fancy?
Answer 1.
We have cited a new paper, which you recommended, as reference No.22 and No.23; ‘The role of serum potassium and sodium levels in the development of postpartum hemorrhage. A retrospective study’ and ‘The role of endoglin and its soluble form in pathogenesis of preeclampsia.’
We have also added a new sentence as follows: The present study also lacks any speculation as for the pathogenesis of preeclampsia, which has substantial effects on fetal growth and placental function [22,23].
We also wanted to discuss maternal gingivitis/periodontitis and preterm delivery. However, those debates would require much more content and raise additional issues, so we could not mention them in the present study.
We appreciate your kind consideration and thoughtful advice for publishing our findings.
June 1, 2022
Sincerely,
Naomi Seo
Department of Obstetrics and Gynecology, Osaka City University Graduate School of Medicine, 1-4-3 Asahimachi Abeno-ku Osaka, Osaka, Japan 545-8585
E-mail: d19mb027@gmail.com
Tel: +81 6 6645 3862
Fax: +81 6 6646 5800